# Emotional Effects in Object Recognition by the Visually Impaired People in Grocery Shopping

**DOI:** 10.3390/s22218442

**Published:** 2022-11-03

**Authors:** Michela Balconi, Carlotta Acconito, Laura Angioletti

**Affiliations:** 1International Research Center for Cognitive Applied Neuroscience (IrcCAN), Catholic University of the Sacred Heart, Largo Gemelli, 1, 20123 Milan, Italy; 2Research Unit in Affective and Social Neuroscience, Department of Psychology, Catholic University of the Sacred Heart, Largo Gemelli, 1, 20123 Milan, Italy

**Keywords:** autonomic activity, emotion, object recognition, visually impaired, grocery shopping

## Abstract

To date, neuroscientific literature on consumption patterns of specific categories of consumers, such as people with disability, is still scarce. This study explored the implicit emotional consumer experience of visually impaired (VI) consumers in-store. A group of VI and a control group explored three different product shelves and manipulated target products during a real supermarket shopping experience. Autonomic (SCL, skin conductance level; SCR, skin conductance response; HR, heart rate; PVA, pulse volume amplitude; BVP, blood volume pulse), behavioural and self-report data were collected in relation to three phases of the in-store shopping experience: (i) identification of a product (recognition accuracy, ACC, and reaction times, RTs); (ii) style of product purchase (predominant sense used for shelf exploration, store spatial representation, and ability to orientate themselves); (iii) consumers experience itself, underlying their emotional experience. In the VI group, higher levels of disorientation, difficulty in finding products, and repeating the route independently were discovered. ACC and RTs also varied by product type. VI also showed significantly higher PVA values compared to the control. For some specific categories (pasta category), PVA correlates negatively with time to recognition and positively with simplicity in finding products in the entire sample. In conclusion, VI emotional and cognitive experience of grocery shopping as stressful and frustrating and has a greater cognitive investment, which is mirrored by the activation of a larger autonomic response compared to the control group. Nevertheless, VI ability to search and recognise a specific product is not so different from people without visual impairment.

## 1. Introduction

Customer experience can be defined as the collection of thoughts, emotions, and attitudes that emerge during decision-making as a result of interactions with other people, things, and the environment, leading to cognitive, emotional, sensory, and behavioural reactions [1], has traditionally been an important marketing research focus.

Both implicit and explicit measures can be gathered and examined in order to understand consumer experience in the round [2,3]. The first, in particular, can be represented by electroencephalography or autonomic measures and provide insight into both cognitive and emotional unconscious processes [4]. Conversely, explicit measures refer to conscious processes that the participant is able to express and are reflected by self-report data collected, for instance, through interviews and questionnaires.

The need to design affordable experiences for everyone must always be considered when thinking about the customer experience in any industry and with any product. An experience, in fact, must be favorable, appealing, and pleasurable for all current and potential customers as well as a limited number of specific users. Consequently, the idea of “inclusive design” has evolved over time in accordance with this line of thinking. Specifically, in 2005 the British Standards Institute defined inclusive design as the development of products and/or services that can be used and accessed by the greatest number of people without the requirement for specialist design or specific adaptations. [5]. In order to enable the use of a particular object, product, or service by as many users as possible, the inclusive design places a strong emphasis on getting to know the target audience in all of its facets, with their needs, desires, and expectations. Failure or misunderstanding can result in a customer’s exclusion from the service or frustration with it, as well as cause issues for the business itself, such as high service fees, expensive rectification, and a poor reputation.

Despite this knowledge, however, businesses frequently deal with customers with disabilities unfairly, denying them the same benefits from the shopping experience as other customers [6]. The neglect of vulnerable customers may be one reason for this deficiency on the part of supermarkets and retailers in general. In 1998, the idea of a “vulnerable consumer” was proposed and denoted the potential existence of vulnerability on multiple levels, including physical, cognitive, motivational, and social [7]. People who are blind or visually impaired (VI) frequently struggle to see signage or product labels in supermarkets because they are poorly contrasted and not in Braille. This is an example of a vulnerability that is frequently ignored. These factors, on their part, make it difficult for customers to find and select the appropriate product, frequently resulting in a negative customer experience that is partially caused by the rudeness of the sales team [8].

With reference to VI people, several previous studies strived to enable accessible shopping in this category through assistive technology, such as electronic travel aids (ETAs) [9,10,11,12]. However, VI users’ experiences (both obstacles and needs) have been mainly explored before through explicit measures, such as video cameras, computational modules (e.g., smartphones or tablet), and surveys [11].

In order to identify and solve, at least in part, the challenges that prevent the creation of a positive and satisfying experience by all consumers, a significant contribution is made by the neuromarketing discipline [3,13], which employs a variety of specific neuroscience tools to record psychophysiological and electrothermal peripheral activity and evaluate heart rate and respiratory rate of body parameters autonomic (skin conductance level and response (SCL and SCR), heart rate (HR), pulse volume amplitude (PVA), and blood volume pulse (BVP)) [14]. These indices, which can be obtained through a biofeedback system, can be used to gather relevant data about the implicit processing that underlies how people interact with stimuli and enable inferences about their emotional and attentional states that are not always possible with traditional self-report questionnaires or scales [15]. However, in this context, there has been more limited empirical research conducted inside a supermarket to detect autonomic and psychophysiological indices in order to study implicit consumer behaviour [16,17]. Next to this aspect, to date, there is a gap in the literature on VI consumers’ behaviour studied with psychophysiological measures. Finally, to the best of our knowledge, no in-store consumer research took place in Italy comparing Italian VI people with sighted consumers’ behaviour.

Given these preconditions, this study consists of behavioural psychophysiological research focused on the products’ choice behaviour inside a supermarket and based on the comparison between VI and sighted people to better highlight the differences in experience in terms of psychological aspects and shopping styles. In this study, video observations, semi-structured interviews, and autonomic measures were exploited to collect behavioural and self-report information.

A new instrument has been developed to identify the most important elements of the purchasing process for the VI persons in order to fill the gap in the literature. This tool is specifically made to gather psychophysiological, behavioural, and self-reporting measures, identifying three key phases of the in-store shopping experience: (i) identification of a product in terms of recognition accuracy and time; (ii) the style of product purchased, in terms of the predominant sense used for shelf exploration, store spatial representation, and ability to orientate themselves; (iii) consumers experience, in terms of self-reported quality of the experience (e.g., perceived stress levels). All these phases are discussed with respect to a particular chosen product category: fruit, pasta, and frozen foods.

Regarding specifics, this paper aims to investigate the autonomic, behavioural, and self-reported measures during the exploration and identification of selected product categories inside a supermarket. In particular, the psychophysiological markers are represented by skin conductance (SC), HR, BVP, and PVA, collected by a biofeedback system. Specifically, these peripheral indices show evidence of both an immediate emotional engagement and a controlled and reasoned cognitive process [18,19]. HR and SC, for example, increase with more painful situations [20], while BVP is an indirect index of vasomotor activity, and PVA is influenced by the heart’s volume, peripheral resistance, vascular diameter as a measure of volume change, and the peripheral vasomotor activity itself [21].

At the behavioural level, it is hypothesised that the VI will exhibit less accurate product recognition and a longer reaction time (RTs) in product identification compared to the control group. That could be because VI requires more cognitive effort to identify a product: they must largely depend on the senses, such as touch or smell, because they are unable to see. Furthermore, it is more difficult to differentiate a product when it is new and undiscovered or when its shape is like that of many other products.

Regarding the self-reported measures, it is assumed that the VI will exhibit lower levels of self-perception in finding products and in the perceived ability to orient themselves independently in the store than the control group. In addition to this, it is also possible to assume for VI higher levels of stress and a greater sense of disorientation as mental confusion in understanding one’s position within the store and the route taken to reach the products. Overall self-reported pleasantness of the in-store experience was expected for both groups, despite any potential problems the VI might discover during shopping.

Concerning autonomic data, firstly, it is expected to observe a higher increase in individuals’ autonomic system responses during the store exploration for the VI people compared to the control group. This could be due to a greater effort that VI people need to identify a product without being able to use sight but using only touch, hearing, and smell. Secondly, this study would like to evaluate the relationship between psychophysiological activity and performance during exploration and identification of products. It is hypothesised to find a link between the psychophysiological index and time to recognise the product, since the activation of a peripheral response mirrors the level of engagement and cognitive resources used [19,22]. Finally, this scientific contribution aims to investigate the link between autonomic activity and self-perception measurements. In this context, it is supposed to observe a link between simplicity in finding products and PVA activation, which represents an index of cognitive effort and emotional arousal [23].

## 2. Materials and Methods

### 2.1. Subsection

A total of 21 participants were recruited for the study by a snowball technique, in order to represent a prototypical sample, and were divided into the VI group ((N = 9 people; 4 male and 5 female; mean age = 47.89; standard deviation = 12.49) and sighted group, named the “control group”, (N = 12 sighted people; 5 male and 7 female; mean age = 24.25; standard deviation = 2.45). The VI group, in particular, was recruited thanks to the “Associazione Nazionale Subvedenti” (ANS) and Milan branch of the Italian Union of the Blind and Visually Impaired, and it included “blind” (N = 5) and “with severe impairment” people (N = 4), on the basis of the degree of visual acuity, as established by Law 138 of 2001.

For participant recruitment, the following exclusion criteria were considered: (a) significant levels of depression evaluated with the Beck Depression Inventory (BDI-II) [24,25] and (b) global cognitive functioning and short- and long-term memory function outside the norm, according to the Mini-Mental State Examination (MMSE) [26,27] and Rey Auditory Verbal Learning Test (RAVLT) [28,29].

Participation in the study was voluntary, without any compensation, and the participants signed a written informed consent form. The research took place in accordance with the Declaration of Helsinki and, furthermore, received the approval of the Ethics Committee of the Department of Psychology, Catholic University of the Sacred Heart, Milan, Italy.

### 2.2. Setting Selection

To identify an adequate and available supermarket for the study, the following parameters were considered: (a) easily reachable with public transport and (b) internal layout with different areas easily recognizable and with a multi-sensory involvement. After a screening of the main large-scale distribution store present in the Milan area in Italy, the choice fell on a well-known supermarket chain.

The experimental sessions took place, in accordance with the store, in a time slot and on less crowded days, and were conducted in three different product areas: fruit, pasta, and frozen foods.

A significant research aspect was that the store exploration represented a new experience for the entire sample since no participant was familiar with the chosen store and, therefore, no one had possible prior spatial maps or knowledge of the layout of products.

### 2.3. Procedure

After the participants arrived at the store, the experimental process began with an assessment step in which psychometric and neuropsychological tests were administered. Then, a non-invasive biofeedback device was placed on the non-dominant hand and recorded psychophysiological activity for 240 s, starting at a neutral wall for 120 s with eyes open and 120 s with eyes closed.

Participants in the experimental in-store phase, characterized by the detection of autonomic activity, were informed that they would have been accompanied to a specific section of the grocery store dedicated to a particular product category (fruits, pasta, frozen food) and that they should have had to explore it for no more than two minutes in order to identify specific target products, that the researchers had previously selected and communicated to them. The specific products selected were a pastina box (short pasta) and a fusilli box (long pasta) for the pasta category, peeled shrimp and scallops for the frozen food category, and orange and grapefruit for the fruit category. These products were chosen because they were placed across several areas distant from each other and were simple to recognize using various senses. The fruit, in fact, is recognizable thanks to touch and smell, pasta with touch and hearing, and frozen food, especially by touch.

During this phase, if the participant was unable to identify the product within the allotted time, the researcher supported them in doing so (this precaution was taken in case there were any issues with the product’s identification so as not to irritate the VI people). Participants were invited to explore the specified product with all their senses before moving on to the next product or area. To provide a behavioral analysis and observe how different participants explored the environment and products, the entire exploration was videotaped.

After the exploration and product identification, each participant was administered a semi-structured interview with the goal of exploring and analyzing the consumers’ experiences in terms of emotional components as well as spatial representation. In order to preserve relevant and significant information, including the precise words the participant used to describe the experience, the interviews were recorded. The duration of the entire experiment was about one hour (see Figure 1).

### 2.4. Behavioural Data Acquisition

Direct nonparticipant observation, in which the researcher observed the events as they were happening but without getting involved in human interaction in the field, was used to collect behavioural data linked to product recognition in the different areas [30]. The quantity of products the participant could identify (accuracy, ACC), as well as reaction times (RTs), were taken into account. Accuracy is, more specifically, the number of products recognised in each exploration area, and RTs is the amount of time it took for each participant to identify the products in each exploration area.

### 2.5. Self-Reported Data Acquisition

Semi-structured interviews were used to gather self-report data with the goal of learning more about the participants’ purchasing and sensory experiences as well as their mental representations of the supermarket in terms of orientation, spatial representation, pleasure, and most used sensory modality. The 27 items in the interview can be divided into four main categories: (i) demographic variables (items 1–7); (ii) consumer experience, which investigated personal experience in terms of stressful level, unpleasantness, and pleasantness (items 8–10); (iii) use of senses (items 11–22); and (iv) orientation ability and spatial representation (items 23–27).

### 2.6. Autonomic Data Acquisition

The autonomic activity was non-invasively collected and recorded using X-pert2000 portable Biofeedback systems with a MULTI radio module (Schuhfried GmbH, Modling, Austria). To record data, a peripheral sensor was placed on the distal phalanx of the second finger of the non-dominant hand. It allows measuring the level and response of SCL, SCR in µS, and HR in beats per minute (bpm). The SCL value was recorded with an EDA gold electrode using current-current measurement at a sampling frequency of 2 kiloHertz (kHz). The use of alternating voltage prevents polarisation. The measurement resolution for the SCL calculation is 12 nanoseconds (ns) with a sampling frequency of 20 Hz. PVA, BVP, and HR were measured via photoplethysmography with a sampling frequency of 500 Hertz (Hz). Furthermore, the mobility of the non-dominant hand was monitored with an accelerometer in meter/square second (m/s^2^) integrated into the sending unit to ensure that the recordings were not compromised by hand movements.

## 3. Results

Mean ± Standard Deviation of behavioral, self-report, and autonomic data for visually impaired and control group are reported in Appendix A section).

### 3.1. Behavioural Data Analysis and Results

An independent-sample t-test (IBM SPSS 25) was applied to the behavioural data (ACC and RTs). The normality of the data distribution was preliminarily assessed and confirmed by checking kurtosis and asymmetry indices. The threshold for statistical significance was set to α = 0.05, absolute t-value for nineteen degrees of freedom (α = 0.05 two tails) corresponds to 2.093. The equality of variances between groups was checked by Levene’s test, which was computed to test the homogeneity of variances between the two groups and to adapt the computation of subsequent inferential tests accordingly. Furthermore, we computed Cohen’s *d* values as a measure of between-group effect size.

The t-test on ACC showed a significant difference between the two groups regarding pasta, with a lower level of ACC for VI group (t = −3.500, p = 0.008, d = −1.798), and regarding frozen food, with a lower level of ACC for VI group (t = −3.162, p = 0.013, d = −1.624). ACC for fruit did not show any significant difference between groups.

The t-test comparing RTs did not show any significant difference between groups regarding any product category.

Additionally, a repeated-measures ANOVA with Group (2: visually impaired, control) as the between-subject factor, and Category (3: fruit, pasta, frozen food) as the within-subject factor was computed for ACC and RTs. Pairwise comparisons were applied to the data in case of significant effects. Simple effects for significant interactions were further checked via pairwise comparisons, and Bonferroni correction was used to reduce multiple comparison potential biases. For all the ANOVA tests, the degrees of freedom were corrected using Greenhouse–Geisser epsilon where appropriate. Furthermore, the normality of the data distribution was preliminarily assessed by checking kurtosis and asymmetry indices. The size of statistically significant effects has been estimated by computing partial eta squared (η^2^) indices.

For the ACC, three significant effects were found: (a) the significant main effect for the between-subject factor Group (F [1,19] = 91.743, p ≤ 0.050, η^2^ = 0.828), where VI exhibited a lower level of ACC compared to control group; (b) the significant main effect for the Category (F [2,38] = 3.800, p = 0.047, η^2^ = 0.167), where higher levels of ACC were found for the fruit’s compared to the pasta recognition; (c) the significant interaction effect Group × Category (F [2,38] = 3.800, p = 0.047, η^2^ = 0.167), where pairwise comparisons revealed a lower level of ACC for VI people compared to control in pasta recognition (p = 0.001) and in frozen food recognition (p = 0.002). In addition, the VI group showed a lower level of ACC in both pasta (p = 0.001) and frozen food recognition (p = 0.032) compared to the fruit category (see Figure 2).

For RTs, a significant interaction effect Group × Category was identified (F [2,38] = 4.220, p = 0.022, η2 = 0.182). Pairwise comparisons showed higher RTs for the VI group compared to the control in the pasta (p = 0.030) and in the frozen food category (p = 0.013).

### 3.2. Self-Reported Data Analysis and Results

Self-reported data were analyzed with the same analysis procedure used for behavioral data and described in the previous section.

The t-test on self-reported data showed a significant difference between the two groups regarding disorientation, with a higher level of disorientation for VI group (t = 2.843, p = 0.018, d = 1.401), about simplicity in finding products, with higher difficulty in the identification for the VI compared to controls (t = −4.001, p = 0.001, d = −1.764), and concerning self-confidence in repeating the route independently, with a lower level of self-confidence for VI group (t = −4.841, p = 0.001, d = −2.451).

### 3.3. Autonomic Data Analysis and Results

A set of mixed repeated measures ANOVAs with Group (2: visually impaired and control) as the between-subject factor and Category (3: fruit, pasta, and frozen food) as the independent within-subject factors was applied on autonomic data.

Pairwise comparisons were applied to the data in case of significant effects. Simple effects for significant interactions were further checked via pairwise comparisons, and Bonferroni correction was used to reduce multiple comparison potential biases. For all the ANOVA tests, the degrees of freedom were corrected using Greenhouse–Geisser epsilon where appropriate. Furthermore, the normality of the data distribution was preliminarily assessed by checking kurtosis and asymmetry indices. The size of statistically significant effects has been estimated by computing partial eta squared (η^2^) indices.

As shown by ANOVA for PVA, a significant main effect in the between-subject factor Group was found (F [1,19] = 15.454, p = 0.001, η^2^= 0.449). Pairwise comparisons revealed significantly higher PVA values for the VI group compared to the control (p = 0.001) (see Figure 2).

No other significant differences were found for the other autonomic data. 

### 3.4. Pearson Correlation between Autonomic and Behavioural Data and Results

Pearson correlations, with Bonferroni corrections for multiple comparisons, were applied between significative autonomic and behavioural data (ACC and RTs). All these correlations were applied firstly on the entire sample, and secondly, participants were divided into two groups.

The statistical analysis showed a statistically significant negative correlation between PVA and RTs (r = −0.451, p = 0.046) for the pasta category in the entire sample (see Figure 3).

No correlational value was significant in every single group.

### 3.5. Pearson Correlation between Autonomic and Self-Reported Data and Results

Pearson correlations, with Bonferroni corrections for multiple comparisons, were also performed between significative autonomic data and the consumers experience dimension (perceived stress level, enjoyable experience level, disorientation, simplicity in finding products, and self-confidence in repeating the route independently).

All these correlations were applied firstly on the entire sample and secondly on participants divided into the two groups.

The statistical analysis showed a statistically significant positive correlation between PVA and simplicity in finding products (r = 0.554, p = 0.011) for the pasta category in the entire sample (see Figure 4).

No correlational value was significant in every single group.

## 4. Discussion

The present study investigated the purchasing behaviour and customer experience of VI people inside a supermarket using explicit and implicit measures. The research was supported by the idea that the shopping experience, and in particular grocery shopping, should be conceived and designed according to the inclusive design model in order to allow a user-friendly experience for all categories of consumers, including those with disability. Multi-measures were considered: autonomic, behavioural (ACC and RTs), and self-reported data (customer experience in terms of perceived stress level, enjoyable experience level, disorientation, simplicity in finding products, and self-confidence in repeating the route independently). Overall, this study can be considered one of the first research exploring psychophysiological correlates of VI people combined with behavioural data in a naturalistic environment (such as the supermarket), and the results discussed below should be considered initial experimental evidence.

In terms of behavioural data, the VI performed less accurately in product recognition than the control group, particularly in the categories of frozen foods and pasta. Overall, fruit was more recognised than pasta, possibly because of its perceptual and sensory characteristics. In terms of RTs, the VI people required more time than the control to recognise pasta and frozen foods. Even though there are no specific studies on VI’s ACC and RTs in product recognition in grocery stores, it is possible to discuss these findings by assuming that numerous aspects are involved in product searching and recognition, including features like the products’ position on the shelf, that may or may not be at eye level [31]. A possible recommendation for marketers and designers could be the rendering of the pasta packages entirely with soft plastic packaging or printing of the superimposition of the shape of the pasta type on the package. While, as far as frozen foods are concerned, given the complexity of the product, the wording of the product in Braille at the point of sale can be one of the most immediate solutions that can be identified, however, it is also necessary to find new ways for helping VI finding and reaching the product within the fridge.

According to self-reported data, VI had more disorientation than the control group and had more difficulty in finding products. In comparison to the control group, the VI also felt less confident about repeating the route independently. This finding is not surprising and supported earlier studies that indicated that indoor navigation, orientation, and product recognition represent the most challenges for the VI inside the supermarket [32,33,34,35,36]. These results add to research arguing that new inclusive and smart solutions are needed to increase the confidence of VI people in finding products and orient themselves in a point of sale.

Requesting supermarket employees to accompany these customers through the store requires an investment of time and personnel. Alternatively, Kulyukin and colleagues (2006) developed a prototype of a robotic shopping assistant for VI consumers [37]. Maike and colleagues (2016) also developed a Universal Navigation, Exploration, and eXchange with Things (U-NEXT) system embedded in a mobile platform to help VI consumer to localise products and obtain product information [38]. Managers can consider investing in these smart solutions that could be beneficial not only for VI people but also for all fragile categories (such as the elderly) that require assistance at the point of sale.

At the autonomic level, a significant increase in PVA was found in the VI group compared to the control. According to the literature, the PVA index provides information about individuals’ emotional arousal and cognitive effort [19,22,23,39,40]. From this point of view, it is possible to explain this result by assuming that the research and the identification of a product is the source of a greater cognitive effort for VI compared to the control. This group of consumers, in fact, could have difficulties in orienting themselves inside the store, in reaching and identifying the desired product, and when they need help, it could occur that the staff is not available or unable to offer the correct support [34,35,36,41]. In all these difficulties, VI people proceed with speculation and verification: they suppose where the desired product is and which pathway to reach it, but also identified it between the shelves after verifying that they are in the correct area of the supermarket [41].

Regarding the correlation between PVA and behavioural data, an interesting result was related to RTs. In the entire sample, longer pasta RTs lead to a decrease in the PVA index. Based on the idea that PVA is also an index of cognitive engagement during the environment’s exploration [19,22,40], it is possible to explain this negative correlation assuming that cognitive engagement is essential to enable products’ identification and recognition. This correlation, in fact, demonstrates that RTs decrease as higher cognitive investment increases, represented by higher PVA values.

Finally, an interesting finding regarding PVA activation and consumer experience (simplicity in finding products) was discovered across the entire group for the pasta category. This particular outcome supports previous findings regarding the relationship between PVA and behavioural data. In fact, a self-reported perception of higher simplicity in finding products correlates with rising PVA levels and cognitive effort index [19,22,40]. It is interesting to note that the implicit autonomic measure supports the explicit self-report measure, demonstrating the value of conducting studies using an approach that incorporates both elements.

All these correlations could also be analysed from an emotional point of view. Even if there are no studies about in-store shopping experience to support this finding, the following papers, in which PVA is examined not just as a cognitive activity but also as an emotional response, suggest that this index measures emotional activation [19,23,39]. Therefore, it is possible to interpret an overall increase in PVA over the entire sample as a growth in the positive emotional reaction that develops in connection with well-known and liked products.

To summarise, the present study permitted exploring the VI people’s behaviour with autonomic, behavioural, and self-reported data during grocery shopping, providing evidence for the importance of including tools for assessing implicit cognitive and emotional processes in addition to those for collecting self-reported data. Thanks to the biofeedback, it was possible to observe a significant activation of the PVA index throughout the store exploration and product identification, particularly pasta.

The use of behavioural data in this study was useful to objectively understand the main difficulties the VI group had when trying to identify specific products, product categories and which product categories compared to others. Collecting both behavioral and self-report data allowed us to obtain a comprehensive overview of VI’s shopping experience, combining objective measures of the difficulties (for instance, through the TR) with subjective measures concerning the self-representation of the individual. Additionally, comparing these two types of information (behavioral and self-report) enables the identification of potential mismatches between actual performance, represented by indirect data such as TRs, and self-evaluation, considered explicit data that is communicated directly by the participant and filtered by the experience. Finally, behavioral data combined with autonomic data can be relevant to evaluating the in-store shopping experience from an emotional perspective.

In conclusion, therefore, the integration of behavioural data with self-report and autonomic data permitted to obtain a wide and detailed knowledge of the consumer experience, investigating the objective and subjective experience, the implicit and explicit level, and both the cognitive and emotional aspects. In contrast to previous approaches used to examine the consumer experience, this research methodology is innovative since it focuses specifically on emotional, implicit, and explicit elements.

These indicators have allowed researchers to highlight that for VI people the shopping experience is a source of higher cognitive engagement, greater emotional arousal, and deeper peripheral resource consumption compared to people with no such disabilities. All these results can be read as indicators of the need for more commitment to developing inclusive design shopping experiences. In particular, the findings from the analysis of behavioural data, when combined with autonomic and self-report data, could be used to develop specific apps or offer instructions for consumers who are blind or VI in order to help them make purchases while feeling as relaxed and supported as possible and to lower their stress levels. With this in mind, it could be advantageous to include support to make identification easier, particularly for those product categories where VI has had more difficulty, like frozen meals and pasta. In this sense, to be truly inclusive, in-store marketing strategies should incorporate tactile touchpoints, braille maps, or an initial guided exploration of the supermarket. Moreover, future studies could test the strength of PVA as physiological markers and evaluate the possibility to use physiological measures to test ETAs.

Despite the project’s inventiveness, it is necessary to be cautious with the interpretation of present results that constitute initial experimental evidence, and some limitations could be addressed. First of all, these findings might be correlate with neurophysiological data in order to obtain further insight into implicit cognitive processes. In the second instance, these findings concern a specific supermarket with its own internal layout of products and, therefore, may not be replicated for another store. It is, in fact, possible that a supermarket chain, different from the one in the present study, has a different internal layout in terms of products and areas explored. Third, it should be addressed that the sample size was low and a probabilistic convenience sampling was adopted for this study. Therefore, the number of participants should be increased in a way to improve greater generalisability of results, as well as age and gender differences should be controlled and balanced in future studies. Perhaps due to the reduced sample size, the statistical effect size we obtained from our results were not large, and this evidence should benefit from replication to be confirmed.

Finally, these findings might be confronted with psychophysiological and neurophysiological data of the fruition phase in order to highlight differences between these two phases.

## Figures and Tables

**Figure 1 sensors-22-08442-f001:**
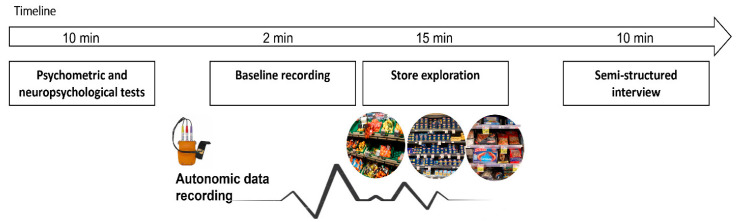
Procedure. Experimental setting with autonomic measures recording during in-store exploration.

**Figure 2 sensors-22-08442-f002:**
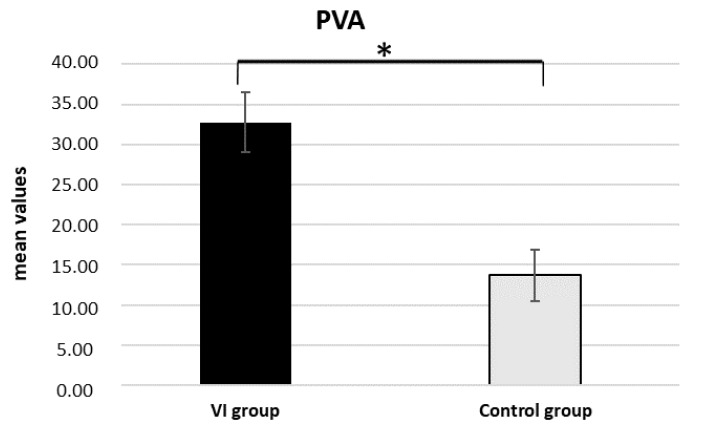
Autonomic result. The bar graph shows significant differences for PVA values between the VI and control group. Bars represent ±1 SE. Star (*) marks statistically significant pairwise comparisons.

**Figure 3 sensors-22-08442-f003:**
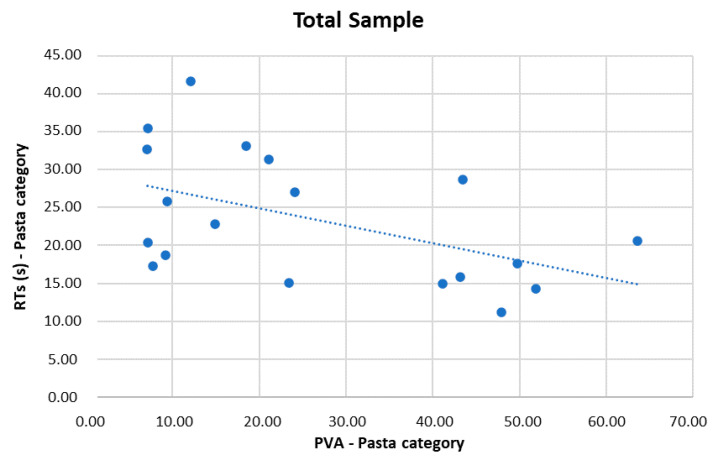
Correlation result: PVA and behavioural data. Significant negative correlation between RTs and PVA values for the entire group in pasta category.

**Figure 4 sensors-22-08442-f004:**
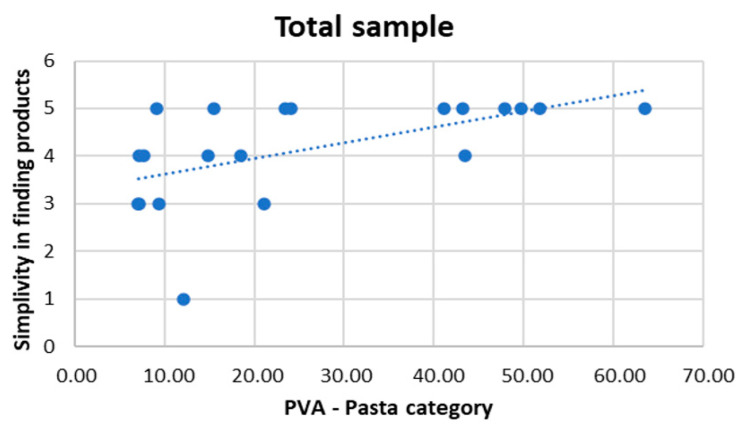
Correlation result: PVA and consumer experience. Significant positive correlation between simplicity in findings products and PVA in the pasta category for the entire group.

## Data Availability

The datasets used and/or analysed during the current study are available from the corresponding author upon reasonable request.

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
