# Peer review of "Emotional Effects in Object Recognition by the Visually Impaired People in Grocery Shopping"

_sensors, 2022, doi:10.3390/s22218442_

Round 1
Reviewer 1 Report
This article focuses on visual impaired people's shopping experience in markets. Through collecting behavior data, this study reveals the difficulties that VI group have. Several issues remain:
- The sample size is quite limited, N=20 in total. The effect size is not high.
- The results reveal the differences but what does it contribute to our understanding? VI people have difficulties for shopping, which we can expect. Maybe, it requires more interpretation of data. For instance, what behavioral data means? What implications results can tell? How should designers and managers do to help IV?
- Some format issues exist. e.g., reports of F test. sentence in line 78-79
Reviewer 2 Report
Dear authors,
The topic is good but I have some suggestions in order to improve your manuscript.
1.the introduction is too general. Try to offer precise data regarding the problem. Lines 77-78 how do you support your affirmation?
2.Your oaper is poor referenced because it lacks a deeper research. Please create a literature review section regarding the issue.
3.Your sample consists 21 oarticipants. It is a proper number? How do justify it?
4.try to improve the discussion section
Round 2
Reviewer 1 Report
I don't think authors understand my previous comment. Let me put it more clearly:
- VI people have difficulty for shopping, VI people have difficulty for searching products, VI people has to repeating their route. All these 'findings' are common sense. In other words, these findings are not novel at all.
- With these 'common sense findings', the novelty of this study (probably) lies in the usage of behavior data to investigate VI people's shopping experience. Then, it is crucial to clarify, why behavior data is important to use? What benefits of using behavior data in comparison to traditional way of data collection? And how the way of using behavioral data, as a novel methodology, can be used for other research and practice?
- More importantly, authors need to clarify how this method of using behavior data can provide implications for the research on VI people and for developing devices for VI people in practice. These are key contributions of this study.
Reviewer 2 Report
Dear authors,
Thank you for your response
